# Evolution of optimal growth temperature in Asgard archaea inferred from the temperature dependence of GDP binding to EF-1A

Zhongyi Lu[1,2,4], Runyue Xia [1,2,4], Siyu Zhang[1,2], Jie Pan[1,2], Yang Liu [1,2], Yuri I. Wolf [3], Eugene V. Koonin [3] & Meng Li [1,2] ✉

The archaeal ancestor of eukaryotes apparently belonged to the phylum *Asgardarchaeota*, but the ecology and evolution of Asgard archaea are poorly understood. The optimal GDP-binding temperature of a translation elongation factor (EF-1A or EF-Tu) has been previously shown to correlate with the optimal growth temperature of diverse prokaryotes. Here, we reconstruct ancestral EF-1A sequences and experimentally measure the optimal GDP-binding temperature of EF-1A from ancient and extant Asgard archaea, to infer the evolution of optimal growth temperatures in *Asgardarchaeota*. Our results suggest that the Asgard ancestor of eukaryotes was a moderate thermophile, with an optimal growth temperature around 53 °C. The origin of eukaryotes appears to coincide with a transition from thermophilic to mesophilic lifestyle during the evolution of Asgard archaea.

Phylogenomic studies of Asgard archaea have clearly shown that the archaeal ancestor of eukaryotes belonged to this archaeal phylum, likely, affiliated with the class *Heimdallarchaeia*[1–4]. Thus, biological and ecological characterization of Asgard archaea is essential for the reconstruction of eukaryogenesis and the environment in which eukaryotes evolved. The optimum growth temperature (OGT) of any microbe is a key physiological trait that typically corresponds to the temperature in its native environment and substantially affects molecular and cellular properties. Thus, exploring the evolution of Asgard OGT can provide insight into the temperature regime of eukaryogenesis.

The elongation factor 1A (EF-1A) in archaea and eukaryotes and its bacterial ortholog EF-Tu are evolutionarily conserved GTPases involved in the delivery of aminoacyl-tRNA to the ribosome during translation[5,6]. The slow divergence rates of the protein sequences of the elongation factors make these proteins useful phylogenetic markers for the analysis of the deepest events in the evolution of life and

ancestral protein reconstruction[5]. The optimal temperature of GDP binding by EF-1A/EF-Tu has been shown to closely correlate with the OGT of a broad range of prokaryotes[5,6]. Therefore, reconstruction of the amino acid sequences of ancestral elongation factors provides a proxy for the inference of the OGT of the corresponding ancestral archaea and bacteria. Here, we characterize the optimal GDP-binding temperature of Asgard EF-1A and the reconstructed ancestral proteins to trace the OGT evolution of these organisms and infer the OGT of the immediate ancestors of eukaryotes.

## Results

### Phylogenetic analysis of Asgard EF-1A

A total of 106 Asgard EF-1A sequences (COG000628) were obtained from our local Asgard archaea database[3]. We analyzed the phylogeny of the EF-1A from Asgard, Euryarchaea, DPANN and TACK archaea, as well as the eukaryotic EF-1A and bacterial EF-Tu using the recommended best-fitting empirical model (LG + F + I + G4) and mixture

[1]Archaeal Biology Center, Institute for Advanced Study, Shenzhen University, Shenzhen 518060, China. [2]Shenzhen Key Laboratory of Marine Microbiome Engineering, Institute for Advanced Study, Shenzhen University, Shenzhen 518060, China. [3]National Center for Biotechnology Information, National Library of Medicine, Bethesda, MD 20894, USA. [4]These authors contributed equally: Zhongyi Lu, Runyue Xia. ✉e-mail: limeng848@szu.edu.cn

model (Q.pfam+C40 + F + R6) in IQ-tree, respectively (see Methods). The two resulting maximum likelihood phylogenetic trees have closely similar overall topologies where the Asgard EF-1A forms a strongly supported clade, with the eukaryotic branch confidently lodged within Asgard which is compatible with a direct ancestral relationship (Supplementary Fig. S1). Specifically, in the empirical model-inferred tree, the eukaryotic clade is the sister group of the *Heimdallarchaeia-Kariarchaeia-Gerdarchaeia-Hodarchaeia* clade, in a better accord with the phylogeny we previously obtained from a concatenation of 29 universally conserved proteins[3]. Therefore, the empirical model-inferred phylogeny was selected for subsequent analyses, in particular, the inference and reconstruction of ancestral forms of EF-1A (Fig. 1a, Supplementary Fig. S2).

## Reconstruction of ancestral EF-1A of the last Asgard ancestor of eukaryotes

To infer the ancestral sequence (AncEF-1A) at the node where the Asgard and eukaryotic EF-1A separated in the phylogeny, we used three independent methods implemented in IQ-tree (v1.6.5, IQ-Anc)[7], FastML (FASTML-Anc)[8], and PAML (v4.9, PAML-Anc)[9]. Despite some amino acid sequence differences in the G-domain, Domain II, and Domain III among the three reconstructions, the resulting ancestral proteins contained identical GDP-binding motifs (motif I, II, and III, Fig. 2a)[10]. We inferred the AncEF-1A sequence based on the consensus sequence of these EF-1A reconstructions and obtained structural models for all reconstructed ancestral sequences proteins using AlphaFold2 through the ColabFold server[11]. The reconstructed AncEF-1A sequence had a mean posterior probability across sites >0.92, which indicates a high confidence in the reconstruction of the EF-1A of the Asgard ancestor of eukaryotes (Fig. 2b). The structures predicted for the three AncEF-1A reconstructions were closely similar to each other and to the consensus model, with RMSD values of 0.156 Å, 0.188 Å and 0.170 Å between the consensus and the PAML-Anc, FastML-Anc, and IQ-Anc reconstructions, respectively (Fig. 2c).

As a control for AncEF-1A, we also modeled EF-1A from *Candidatus* Prometheoarchaeum syntrophicum MK-D1 strain, the first cultured

Asgard archaeon that grows optimally at around 20 °C[2]. The structural model of AncEF-1A was found to be closely similar to that of MK-D1 EF-1A (RSMD = 0.520 Å, Fig. 1b), both adopting the typical EF structure containing the conserved GDP-binding residues within the G-domain. To ascertain the correspondence between the optimal GDP-binding temperature of EF-1A and the organism's OGT, we determined the thermal stability of MK-D1 EF-1A across 0 to 100 °C using ThermoFluor assay (Supplementary Fig. S3), and then measured the temperature dependency of Mant-GDP binding across the stable temperature range as previously described[5]. The MK-D1 EF-1A was stable across the temperature range from 0 to 60 °C and showed optimal GDP binding at about 20.5 °C (90% confidence interval (CI): 19–24.1 °C; Fig. 1c, Supplementary Fig. S4, and Supplementary Data 1) which coincides with the OGT of this organism. The temperature of 60 °C perhaps can be considered an approximate upper temperature limit for MK-D1. In addition, we performed the same assay for EF-Tu of *Escherichia coli* and EF-1A proteins of *Saccharomyces cerevisiae* and *Homo sapiens*. The temperature optima for GDP binding were 40.2 °C for *E. coli* (CI: 37–43 °C), 34.6 °C for *S. cerevisiae* (CI: 33.2–36.3 °C), and 40.1 °C for *H. sapiens* (CI: 37.1–42.9 °C), close to the optimal temperature for physiological activities of these organisms (Pearson correlation coefficient r = 0.9906; Supplementary Figs. S4, S5, and Supplementary Data 1). Thus, all these control experiments validate the strong correlation between the temperature optimum- for Mant-GDP binding by EF-1A and OGT.

To characterize the optimal GDP-binding temperature of AncEF-1A, we codon- optimized the coding sequence of this protein in *E. coli* BL21, and assembled the coding sequence into the pCold-II vector. We expressed and purified AncEF-1A and determined its thermal stability in the range of 0–70 °C (Supplementary Fig. S2) and the temperature dependence of GDP binding, observing the optimum at 52.9 °C (90% CI: 51.2–55.6 °C; Fig. 1c, Supplementary Fig. S4, and Supplementary Data 1). In addition, we performed the same test for IQ-Anc protein and observed the optimal GDP binding at 49.4 °C (90% CI: 45.8–50.1 °C; Supplementary Figs. S3–S5, and Supplementary Data 1). These findings make it highly unlikely that a small number of amino acid residue

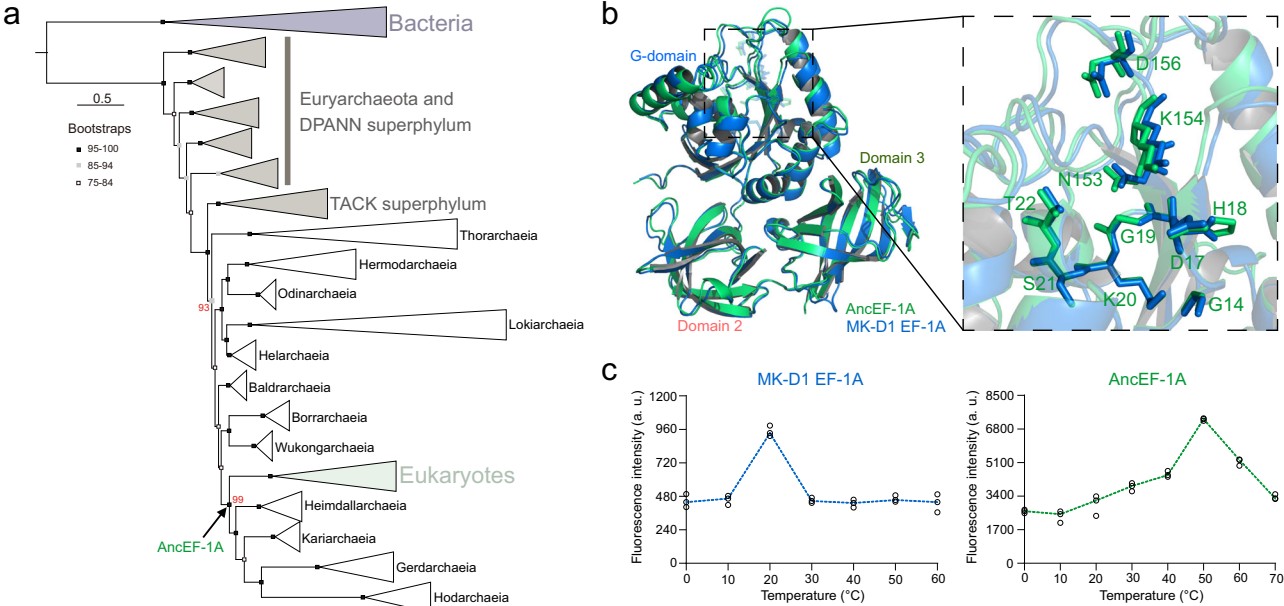

**Fig. 1 | Origin of eukaryotic EF-1A within Asgard archaea. a** Simplified maximum likelihood phylogeny of archaeal and eukaryotic EF-1A and bacterial EF-Tu (used as an outgroup). Bootstrap values are shown for nodes at which the common ancestry of Asgard EF-1A might emerge, and the eukaryotic and Asgard EF-1A separated, respectively. **b** A superimposition of the AncEF-1A (green) and the MK-D1 EF-1A (blue). The key GDP-binding residues in motif I and motif II of the AncEF-1A are shown. **c** Evaluation of the GDP-binding capability of the AncEF-1A and MK-D1 EF-1A at different temperature conditions. a. u., arbitrary units. Source data are provided as a Source Data file.

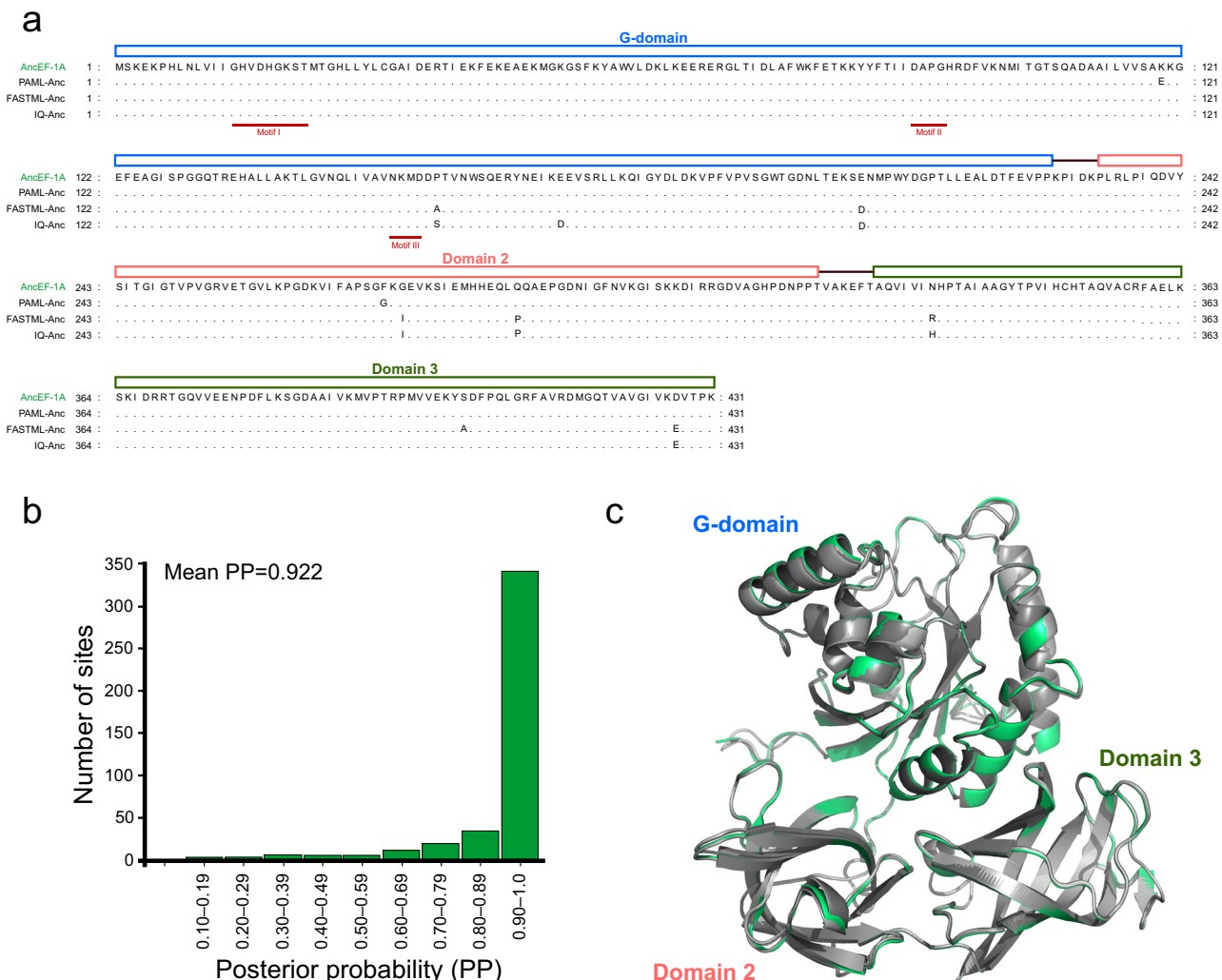

**Fig. 2 | Amino acid sequence and structural characterization of AncEF-1A.**
**a** Amino acid sequence alignment of AncEF-1A and its reference ancestral EF-1A proteins that were reconstructed by PAML (PAML-Anc), FastML (FastML-Anc), and IQ-tree (IQ-Anc), respectively. The dots indicate the states that are identical to AncEF-1A. The proposed G-domain, Domain 2, and Domain 3 of the AncEF-1A, and its GDP-binding residues in motif I and motif II are shown. **b** Distribution of Posterior probabilities for ancestral amino acid states of the AncEF-1A. **c** A superimposition of the AncEF-1A (green) and its reference ancestral EF-1A proteins (gray). The proposed G-domain, Domain 2, and Domain 3 of the AncEF-1A are shown.

changes substantially affects GDP binding by EF-1A. Given the strong correlation between the temperature optimum for GDP binding and OGT, these results suggest that the Asgard ancestor of eukaryotes was a moderate thermophile, with OGT around 53 °C. Notably, this conclusion agrees with the inferred OGT for the alphaproteobacterial ancestor of the mitochondria (51–53 °C)[6]. Combining these findings with the numerous observations that put the upper thermal limit for eukaryotes below 60 °C[12,13], it appears most likely that eukaryogenesis occurred at around 50 °C, which is close to the temperature of bacterial mats in hot springs[12,13].

### Analysis of the OGT evolution in *Asgardarchaeota*

To explore the evolution of OGT in *Asgardarchaeota*, we expressed EF-1A from representatives of each Asgard class except for *Lokiarchaeia*, which was split into two subgroups based on the comparatively low sequence similarity (63.4% identity vs >74% average identity within the other classes; Supplementary Fig. S6), and tested their stability temperature range before measuring the optimal GDP-binding temperature (Supplementary Fig. S3). Taking this temperature as a proxy for OGT, our results suggest that *Baldrarchaeia* (64 °C, 90% CI:

61.4–69 °C), *Odinarchaeia* (57.2 °C, 90% CI: 55.9–58.8 °C), *Helarchaeia* (61.1 °C, 90% CI: 60.2–63.2 °C), *Wukongarchaeia* (63.2 °C, 90% CI: 54.8–65.6 °C), and *Borrarchaeia* (49.5 °C, 90% CI: 49.4–49.6 °C) are moderate thermophiles, whereas *Thorarchaeia* (33.8 °C, 90% CI 33.5–34 °C), *Hermodarchaeia* (40.0 °C, 90% CI: 40.0–40.0 °C), *Heimdallarchaeia* (28.6 °C, 90% CI: 24.8–33.1 °C), *Kariarchaeia* (30.3 °C, 90% CI: 27.6–32.6 °C), *Hodarchaeia* (40 °C, 90% CI: 40.0–40.0 °C), and *Gerdarchaeia* (32.8 °C, 90% CI: 30.9–33.1 °C) are mesophiles (Fig. 3, Supplementary Figs. S4, S5, and Supplementary Data 1). The *Lokiarchaeia* ancestor (57.5 °C, 90% CI: 56.1–60.2 °C) seems to split into the mesophilic group Loki-2 (MK-D1; 20.5 °C; 90% CI: 19–24.1 °C) and the thermophilic group Loki-3 (50.1 °C, 90% CI: 49.5–50.7 °C) for which different lifestyles have been documented (Supplementary Figs. S4 and S5)[2,14].

We additionally predicted the OGTs across the Asgard diversity based on genomic features (see Methods, Supplementary Data 2)[15]. The predicted Asgard OGTs showed a strong correlation (Pearson correlation coefficient r = 0.7697, Supplementary Fig. S7a) with those inferred by EF-1A GDP-binding temperature. Because a recent study has predicted the Asgard OGTs by an alternative method based on

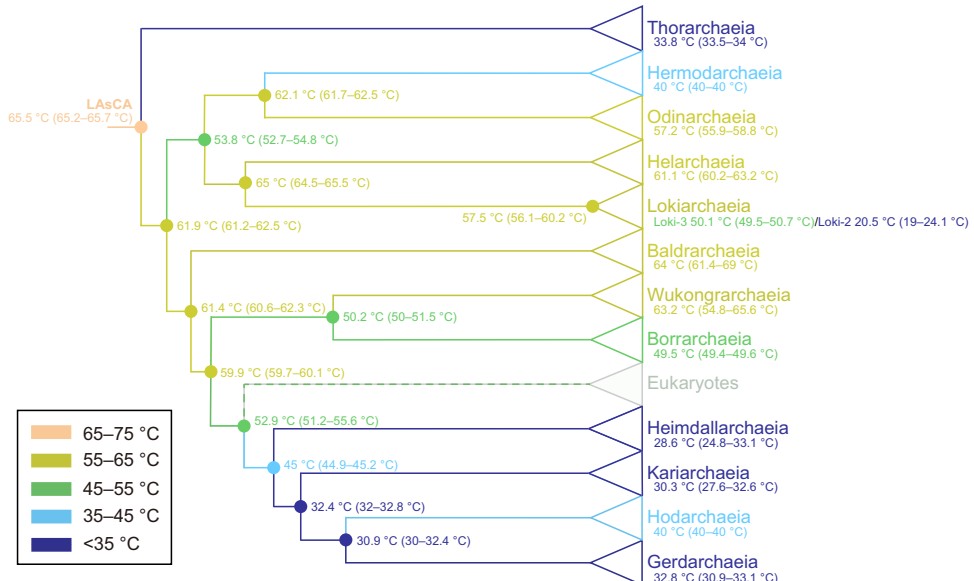

**Fig. 3 | Divergent evolution of OGTs across the Asgard archaea.** Evolutionary trajectory of Asgard OGTs is based on evolution of EF-1A proteins. Branches and nodes are labeled and colored according to the optimal GDP-binding temperature of the Asgard and ancestral EF-1A proteins. Source data are provided as a Source Data file.

genomic features[4], we also performed a correlation analysis between our GDP-binding data and their inferred Asgard OGTs, and observed a moderate correlation (Pearson correlation coefficient r = 0.6289, Supplementary Fig. S7b). Our OGT inference based on genomic features yielded values close to those inferred from GDP-binding by EF-1A for the majority of asgardarchaeal classes (Supplementary Data 2). However, several discrepancies were observed as well. In particular, the prediction from genomic features gave the OGT of 31.6 °C for MK-D1 in the mesophilic Loki-2 group (OGT median 34.2 °C); the Loki-3 group was found to be mesophilic (OGT median 40.5 °C; Supplementary Data 2); *Baldrarchaeia* were inferred to be hyperthermophiles (OGT median 84.9 °C) whereas *Helarchaeia* (OGT median 43.2 °C), and *Borrarchaeia* (OGT median 47.4 °C) came out as mesophiles. These differences between the OGTs inferred with the two approaches for a minority of the asgardarchaeal classes might stem from biases in genomic features such as those caused by extensive horizontal gene transfer[16,17].

We next reconstructed the ancestral Asgard EF-1A proteins at nodes across the phylogenetic tree of Asgard archaea based on the three independent algorithms of IQ-tree, FastML, and PAML, and determined their optimal GDP-binding temperatures, in order to trace the Asgard OGT evolution. The sequences of these ancestral EF-1A proteins were inferred with high confidence (mean PP > 0.8; Supplementary Fig. S8; Supplementary Data 3), and were codon optimized, synthesized, and expressed in *E. coli* BL21. We purified these reconstructed ancestral proteins, evaluated their thermal stability range (Supplementary Fig. S3), and determined their optimal GDP-binding temperatures. The results suggested that Asgard archaea descended from a thermophilic Last Asgard Common Ancestor (LAsCA, OGT 65.5 °C, 90% CI: 65.2–65.7 °C; Fig. 3, Supplementary Figs. S4, S5, and Supplementary Data 1). The general trend in the subsequent evolution apparently was towards decreasing OGT, and the branch leading to the eukaryotes, in particular, split from a common ancestor with the mesophilic branch of Asgard archaea consisting of *Heimdallarchaeia*, *Kariarchaeia*, *Hodarchaeia*, and *Gerdarchaeia* (Fig. 3). Thus, the origin of eukaryotes appears to coincide with the transition from the moderate thermophilic to the mesophilic lifestyle in the evolution of Asgard archaea (Fig. 3).

## Discussion

Our work reveals the evolutionary trajectory of OGT from the common ancestor of Asgardarchaeota to its extant descendants through the evolutionary intermediates, based on the optimal GDP-binding temperature of EF-1A. In particular, characterization of the reconstructed EF-1A of the common ancestor of the Asgard clade that includes *Heimdallarchaeia*, *Kariarchaeia*, *Hodarchaeia*, and *Gerdarchaeia* along with eukaryotes suggested that the Asgard ancestor of eukaryotes was a moderate thermophile with an OGT of about 53 °C.

Based on our determination of the optimal temperatures of GDP binding by ancestral EF-1A combined with the inference of the OGT of the alphaproteobacterial ancestor of the mitochondria[6] and the upper temperature limit for eukaryotic life[12,13], we hypothesize that the temperature for eukaryogenesis was about 50 °C. However, because eukaryogenesis was a highly complex evolutionary process, the temperature regime during the transition from the Asgard ancestor of eukaryotes to the Last Eukaryotic Common Ancestor (LECA) might have been influenced by various factors, in particular, the one or more endosymbiotic events that occurred during that stage of evolution[18–26]. Furthermore, the increased oxygen solubility at the moderate temperature could be important for the mitochondrial endosymbiosis given that the mitochondrial ancestor is thought to have possessed the aerobic respiratory capability[19,27]. It should be emphasized that in this work we estimate the OGT of the last Asgard ancestor of eukaryotes (phylogenetically defined as the archaeal lineage positioned between eukaryotes and its closest sister Asgard group) rather than that of the LECA. Although the upper temperature limit for eukaryotic life (about 60 °C), most likely, applies to all stages of eukaryogenesis, the OGT might have shifted towards mesophilic values during this process, and LECA, in particular, was likely a mesophile. Further exploration of the exact ancestry of eukaryotes within the Asgard phylogeny[1,3,4,28] as well as the phylogenetic placement of the alphaproteobacterial ancestor of the mitochondrion[29,30], accompanied by reconstruction of ancestral states, can be expected to clarify the environmental regime of eukaryogenesis.

A notable trend revealed in this work is the gradual decrease of OGT during Asgard evolution. The divergence of the eukaryotic lineage from the Asgard sister group seems to have occurred at a time

when the archaeal ancestors were transitioning from thermophily to mesophily. Our findings are generally compatible with the results of the recent study that predicted the Asgard OGT from genomic features, in particular, the GC content[4], which we also reproduced with our own analysis of genomic features. Given the possible impact of metagenome-assembled genome quality as well as various unrecognized biases in the genomic features on such inferences, phylogenetic and biochemical characterization of EF-1A provides an independent, complementary approach that enables tracing the evolution of the Asgard growth temperatures[31].

Despite the overall strong correlation between the experimentally observed OGT and the OGT inferred from GDP-binding by EF-1A, there are some discrepancies between the latter (as well as the OGT inferred from genome, Supplementary Data 2) and the corresponding sample collection temperature. For instance, MK-D1 was originally isolated from ocean sediment at about 2 °C which is much lower than the OGTs predicted by GDP-binding data (20 °C) and genomic features (30 °C)[2]. Prokaryotes dwell in highly dynamic, complex ecological niches and microbial communities. They have been documented to develop adaptive strategies for surviving across wide temperature ranges, including both heat-shock and cold-shock responses[32–35]. Furthermore, given that ancestral proteins generally tend to be slightly more stable than extant ones[36], some biases might exist in the ancestral OGT inference from GDP-binding. Thus, the OGT inferred from GDP-binding data cannot prescribe the exact environmental temperature condition for Asgard ancestors that likely could thrive within a range of temperatures. Further physiological characterization of the microbes and in situ studies are certainly required to explore the ecology of Asgard archaea across diverse environments.

From a more general planetary evolution perspective, it seems likely that the overall decreasing growth temperature across the Asgard phylum was caused by the progressively cooling palaeo-temperature trend[6,37]. It has been proposed that the emergence of eukaryotes was facilitated by the Great Oxygenation Event (OGE, 2.33 billion years ago) that occurred during the Huronian Glaciation era (2.45–2.2 billion years ago)[19,38–41]. Glaciation likely caused a general drop in paleoenvironmental temperatures, creating cooler niches that were conducive to eukaryogenesis. The insight into the evolutionary trajectory of the growth temperature across Asgard archaea reported here should guide physiological and ecological study of this group of organisms, shedding light on eukaryogenesis.

## Methods

### Phylogenetic analysis
A total of 195 EF-1A and EF-Tu protein sequences (including 106 Asgard EF-1A sequences) were obtained by UniProt database (https://www.uniprot.org/), NCBI database (https://www.ncbi.nlm.nih.gov/), and our local Asgard archaea database[3]. These sequences were aligned using MUSCLE (v3.8.1551)[42] and their distinct N- and C-terminal regions were manually trimmed (the trimmed sequences are shown in Fig. S6) prior to the construction of a maximum likelihood phylogenetic tree using IQ-Tree (v1.6.5)[7]. The tree was constructed using either the recommended best-fitting empirical model LG + F + I + G4 or the mixture model Q.pfam+C40 + F + R6 in IQ-tree[7] (Supplementary Fig. S1).

### Reconstruction of ancestral sequences
To infer the ancestral sequences from nodes of the EF-1A/Tu maximum likelihood phylogenetic tree with minimal ambiguity, three independent methods implemented in IQ-tree (v1.6.5)[7], FastML[8], and PAML (v4.9)[9] were used here. The ancestral state of nodes in the EF-1A/Tu phylogenetic tree reconstructed was empirical Bayesian method, with option "-asr", in IQ-tree (v1.6.5). Based on the phylogenetic tree, the ancestral amino acid states of protein from each node were then inferred using the FastML web server with the WAG substitution

model. Additionally, the ancestral sequence reconstruction was performed using the AAML module in PAML (v4.9) with an empirical model and wag. DAT substitution rate matrix. The generated ancestral sequences were aligned using MUSCLE (v3.8.1551), and manually corrected given the constraint of neighbor EF-1A sequences when inconsistent amino acid states or gaps occurred. The posterior probability (PP) and mean PP for the resulting ancestral sequences were calculated using IQ-tree (v1.6.5).

### In silico protein modeling
The EF-1A structure model was built using the ColabFold server that employs the AlphaFold2 method for protein structure prediction, with the default parameters (multiple sequence alignment model: UniRef +Environmental, model type: auto, pair mode: unpaired+paired, number of recycle: 3)[11]. The resulting protein structure with the highest lDDT score was used here.

### Protein expression and purification
The coding sequence of EF-1A was codon optimized and synthesized (General Biol Inc., China), and cloned into the pCold-II or pCold-TF vectors (TaKaRa Bio Inc., Japan). The pCold-II has a N-terminal His tag, whereas the pCold-TF contains a N-terminal His tag and a soluble trigger factor chaperone tag (TF) that contributes the solution of target protein. These resulting recombinant vectors were then transformed into Escherichia coli BL21 (TaKaRa Bio Inc., Japan) for protein expression. Briefly, the E. coli BL21 that bears the recombinant vector were inoculated in Luria-Bertani liquid medium containing 100 µg/mL, and incubated at 37 °C until the $OD_{600}$ reached 0.6, and were induced with 0.2 mM isopropyl β-D-1-thiogalactopyranodside at 15 °C for 16–18 h. The target EF-1A protein was purified by HisTrap HP column in ÄKATA Pure Protein Purification System (GE Healthcare Inc., USA). For protein that was expressed in pCold-TF vector, the TF tag was removed by HRV 3 C Protease (TaKaRa Bio Inc., Japan). All the purified EF-1A proteins were concerned by 10 K AIMCO Ultra-15 (Millipore Inc., USA), and were determined by Bradford Protein Assay Kit (Byotime Bio Inc., China).

### Thermal stability assay
The thermal stability range of each EF-1A protein was determined by ThermoFluor assay with slight modification[43]. The ThermoFluor reaction (50 µL) contains 5 µM EF-1A protein, 1×SYPRO Orange dye (Life Technologies, USA), 20 mM Tris-HCl (pH 7.5), 50 mM KCl, 5 mM $MgCl_2$, and 1 mM dithiothreitol. The reactions were incubated for 5 min at 10 °C intervals in the temperature range from 0 °C to 100 °C. Then, the fluorescence intensity of each reaction was measured at excitation 300 nm and emission 470 nm in Infinite 200 PRO (TECAN Inc., Switzerland). Three independent replicates were performed and the highest fluorescence intensity means that the transition of protein from partial unfolding to compete unfolding occurred.

### GDP binding assay
To determine the temperature dependence of GDP binding by EF-1A protein, binding of Mant-GDP (Thermo Fisher Scientific Inc., USA) was measured at 10 °C intervals in the temperature range from 0 °C to100 °C[5]. The GDP-binding reaction (100 µL) contains 20 mM Tris-HCl (pH 7.5), 50 mM KCl, 5 mM $MgCl_2$, 5 µM Mant-GDP, 1 mM dithiothreitol, and the EF-1A protein (<1 µM). The reactions were incubated for 5 min at different temperatures, and their containing unbound Mant-GDP was removed with MicroSpin G-25 Columns (Cytiva Inc., USA). The fluorescence intensity of the resulting reaction buffer was monitored at excitation 360 nm and emission 460 nm in Infinite 200 PRO (TECAN Inc., Switzerland). Three independent replicates were performed to confirm the trend of fluorescence intensity profile of each EF-1A protein.

## Statistical analysis of the optimal GDP-binding temperature

The 33 fluorescence datapoints (3 independent measurements at 11 temperatures) were fit to a skewed Gaussian-like bell-shaped function

$$F' = F_0 + (F^* - F_0)e^{-\left(\frac{t-t^*}{s(t)}\right)^2} \qquad (1)$$

where $F'$ is the predicted fluorescence, $F_0$ is the baseline fluorescence, $F^*$ is the peak fluorescence, $t$ is the temperature, $t^*$ is the peak temperature, and

$$s(t) = s_L + (s_R - s_L)/(1 + e^{-\frac{t-t^*}{w}}) \qquad (2)$$

is the asymmetric peak width that changes from $s_L$ to $s_R$, around the peak temperature $t^*$ at rate $w = 5$ (i.e., over 5 °C). The least-squares fit for the five parameters ($F_0$, $F^*$, $t^*$, $s_L$ and $s_R$) was obtained using the *optim()* function of the R package.

To obtain the confidence intervals for the peak temperatures, we constructed 1000 bootstrap datasets where, for each temperature point, three out of three fluorescence values were sampled with return. The peak temperature was obtained for each of the bootstrap samples using the procedure described above and the 90% confidence interval was found from the 5th and the 95th percentiles of the peak temperature distribution.

## Statistical analysis of the Pearson correlation coefficient

The Pearson matrix correlation coefficient between the OGT value (independent variable) and EF-1A/Tu optimal GDP-binding temperature (dependent variable) was calculated using Grad prism statistical software (9.0.0). The calculation relied on data of MK-D1 (OGT: 20 °C, optimal GDP-binding temperature: 19.9 °C), *S. cerevisiae* (OGT: 28 °C, optimal GDP-binding temperature: 32.2 °C), *H. sapiens* (OGT: 37 °C, optimal GDP-binding temperature: 33.5 °C), and *E. coli* (OGT: 37 °C, optimal GDP-binding temperature: 43.4 °C). A same analysis of Pearson correlation coefficient was also performed for comparison of the median values of Asgard OGT inferred from genomic features with those inferred from EF-1A GDP-binding temperature.

## Prediction of optimal growth temperature

The machine learning model Tome (v1.0) was used to predict OGT values of the Asgard archaea based on the proteomic features[15]. The computer code can be found in https://github.com/EngqvistLab/Tome.

## Reporting summary

Further information on research design is available in the Nature Portfolio Reporting Summary linked to this article.

# Data availability

The bioinformatic and biochemical data generated in this study are available in the Supplementary Information and Source Data files. Source data are provided with this paper.

# Code availability

The script to run the statistical analysis of the optimal GDP-binding temperature is available at Zenodo (https://doi.org/10.5281/zenodo.8433221).

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

## Acknowledgements
This work was supported by the National Natural Science Foundation of China (No. 32225003, 32393970, 32393971, 92251306 to M.L., 32370004, 32000002 to Z.L., 92051102 to Y.L.), Guangdong Basic and Applied Basic Research Foundation (grant no. 2023A1515011309 to Z.L.), the Shenzhen Medical Research Funding (No. B2301005 to M.L.), the Innovation Team Project of Universities in Guangdong Province (No. 2020KCXTD023 to M.L.) and Shenzhen University 2035 Program for Excellent Research (No. 2022B002 to M.L.). Y.I.W. and E.V.K. are supported by the Intramural Research Program of the National Institutes of Health of the USA (National Library of Medicine).

## Author contributions
Z.L., R.X. and M.L. conceived and designed the experiments. R.X. and S.Z. performed the experiments. Z.L., S.Z., J.P. and Y.L. performed the bioinformatics analyses. Z.L., R.X. and Y.I.W. analyzed the data. Z.L., R.X., E.V.K. and M.L. wrote the paper, and all authors edited and approved the paper.

## Competing interests
The authors declare no competing interests.
