## [Peer Review File · Nature Communications]

Evolution of optimal growth temperature in Asgard archaea
inferred from the temperature dependence of GDP binding to
EF-1AReviewer #1 (Remarks to the Author):

Lu and colleagues apply ancestral sequence reconstruction to EF ancestors in order to infer the ancient growth temperatures during Asgard Archaea evolution. The authors resurrect many ancient EFs at various nodes of their phylogeny to conclude that the Asgard ancestor of eukaryotes was a moderate thermophile. The authors also characterize many extant EFs from Asgard to better understand growth temperature throughout the lineage. The manuscript presents interesting results that build off previous studies that describe ancient microbial life and thus makes a strong contribution despite the obvious contentious placement of eukaryotes within the Archaeal domain of life (a debate best left to other manuscripts). I am generally enthusiastic about the manuscript but this could be enhanced with additional clarity.

1) The title is a little misleading because optimal growth is not being directly measured. Maybe remove 'optimal'.

2) The authors nicely describe how the ancient AncEF-1alpha sequence was inferred using multiple algorithms and then taking the consensus among those algorithm outputs. It is less clear if the ancestral Asgard nodes were inferred in a similar manner, or whether just one algorithm was used.

3) Figure 3 needs the nodes added from Figure S3. Please add filled-circles to all the nodes in Figure 3 in which an ancient protein was experimentally resurrected and characterized.

4) The authors mention that their results align with the results of a previous study that characterized ancient EF from ancestral alpha-proteobacteria. The authors also briefly mention this a limitation of dissolving molecular oxygen. I think readers would greatly benefit from an expanded discussion of these overlapping results and possible explanations for more complex life at this temperature transition.

Minor points:

- Please add page numbers (always use page numbers).
- P6, L120: 'with to the' is incorrect.
- Ignorant point - I thought people started using eEF1A instead of EF-1alpha
- Some of the temperature profiles in the Supp Material show broad profiles (and sometimes what appears to be a dual temperature optimum). Please discuss these unusual results. Experimental fluctuations, or possible properties from proteins that optimally function across a broad range of temperatures.

Reviewer #2 (Remarks to the Author):

Evolution of optimum growth temperature in Asgard archaea – Lu et al.

In this manuscript Lu et al. measure the binding capacity of Mant-GDP to EF1 α proteins from extant Asgard species and from the inferred protein sequences of lost ancestors. They conclude that the last common ancestor of Asgard archaea had a preferred growth temperature of 65 °C and the Asgard ancestor of eukaryotes favored 50 °C. Such information is important for understanding the emergence of eukaryotes. However, there are a number of important issues to be addressed, particularly with the biochemical analysis, before publication should be considered.

1) The assay uses Mant-GDP to infer the optimal binding of GDP. A control sample is needed that verifies that Mant-GDP is an accurate reporter of GDP binding to EF1 α . To do this the authors can use an EF1 α from a bacterial species reported in reference 4 and relate the optimal binding to that publication. A second EF1 α control from a warm-blooded animal is needed to assess the relationship between Mant-GDP and a known EF1 α native environment. It would also be good to measure the optimal binding temperature for a second sequence from Fig. 2a, to demonstrate that small numbers of amino acid changes do not affect binding.

2) The assay in its current form is flawed. Most proteins denature below 100 °C. It is meaningless to analyze binding of Mant-GDP to denatured protein. Using these data points invalidate the statistical analysis. The authors should evaluate the stability range of each protein and only use/plot Mant-GDP data points that lie within the stable range. The ThermoFluor assay is one choice to measure the stability.

3) The experimentally determined data points should replace the calculated plots in Fig. 1c and Fig. S3b,c. The current versions of these figures can be placed in the SD.

4) The relationships of the optimal binding temperatures to the reported sample collection environments for metagenomics need to be included for all extant species in this analysis. There seems to be large discrepancies, which bring the hypothesis into question. MKD1 was isolated from ocean sediment ~2 °C. The first Thorarchaeaia and Hermodarchaeaia from estuary and mangrove sediments, respectively. It is difficult to reconcile these environments with the optimal binding temperatures from the Mant-GDP assay of 23, 94, 85 °C, respectively. A thorough discussion of this issue is required.

5) The optimal binding temperature should be written on the branches and nodes in Fig. 3 and Fig S3a to aid the readers' understanding of the results.

Reviewer #3 (Remarks to the Author):

This manuscript by Lu et al. addresses the evolution of optimal growth temperatures across the Asgard archaea. To investigate this, the authors focus on the elongation factor 1alpha (EF-1alpha), a crucial component required for protein synthesis by the ribosome, whose affinity for GDP is optimal at a temperature that approximates that of optimal organismal growth. They reconstruct ancestral sequences for EF-1alpha across the phylogeny of the Archaea and experimentally test the optimal GDP-binding temperature of these ancestral sequences expressed in and purified from *E. coli*. The authors derive at least two broad conclusions: (1) there is somewhat of an overall macroevolutionary trend towards a decrease in OGT in Asgard archaea, and (2) the divergence of the eukaryotic nucleocytoplasmic lineage from Asgard archaea occurred at a point where these archaeal ancestors had become more mesophilic—the archaeal ancestor of eukaryotes was slightly thermophilic by being adapted to 50-60C.

Although this is an interesting study that addresses an important question, I have some methodological concerns and would caution against the broad conclusions that are stated. I thus think that this manuscript has to undergo major revisions before it can be published in a journal of broad readership.

It is most likely that the model used for phylogenetic inference and ancestral sequence reconstruction is not the best-fitting model as it does not account for cross-site heterogeneity in amino acid composition. IQ-TREE includes a set of mixture models (e.g., C10-60) that are particularly suited to deep phylogenetic issues like the one addressed here. The 'TESTONLY' option used by the authors would not test mixture models prior to phylogenetic analysis. This is a rather unfortunate choice of parameters.

What is the effect of trimming on ancestral sequence reconstruction? Depending on the quality of the alignment and the stringency of the trimming, it could have serious effects. The authors need to explore this or at least provide a proper explanation.

The authors should also estimate the optimal growth temperature of the genomes analyzed using the genome-wide methods reported previously (<https://doi.org/10.1093/bioinformatics/btz059>). Ancestral reconstruction could then be made on these estimates using comparative phylogenetic methods, e.g., using models of continuous character evolution. These ancestral estimates should

be compared to those obtained in this study. This is an important analysis because it constitutes an independent approach that may add or decrease support to their conclusions.

What is the precise relationship between optimal growth and GDP-binding temperatures across species? A scatter plot showing this correlation should be presented. Conclusions about point estimates should incorporate the errors derived from such cross-species phylogenetic correlations.

To properly provide insights into the temperature regime of eukaryogenesis, as the authors claim, the OGT of LECA has to be estimated as well—the authors only address the OGT of FECA in their study. This would provide a temperature range that would better constrain eukaryogenesis. However, for this to be properly done, the taxon sampling for eukaryotes in this study has to be improved considerably. The current taxon sampling is limited to a few animal, fungal, and plant species, in addition to *Trypanosoma brucei*. Please keep in mind that most of eukaryotic diversity is microbial and is currently distributed among several so-called 'supergroups'.

Conclusions about ancestral OGTs chronologically matching the Huronian glaciation era are rather unconvincing in the absence of properly calibrated chronograms (at minimum add time to Fig. 3). The argument is barely elaborated upon. This argument at least deserves a proper discussion of the literature.

What do the authors mean by the following sentence? "The dissolved oxygen at this temperature condition could be necessary for the mitochondrial endosymbiosis occurred since the mitochondrial ancestor is thought to possess aerobic respiratory capability." The meaning and implications of this statement are not clear at all. In fact, dissolved oxygen concentration increases with lower temperatures.

Claims about temperature of mitochondrial ancestor needs to be critically revisited. Obvious biases are present in the study cited by the authors. The authors should probably estimate the OGT of the mitochondrial ancestor using the same methods employed here for Asgard archaea. That would add further support to the optimal temperature range at which eukaryogenesis occurred. Be sure to use the most up-to-date studied on the phylogenetic placement of the mitochondrial ancestor: <https://doi.org/10.1038/s41559-021-01638-2> and <https://doi.org/10.1038/s41586-018-0059-5>

REVIEWER COMMENTS

Reviewer #1 (Remarks to the Author):

Lu and colleagues apply ancestral sequence reconstruction to EF ancestors in order to infer the ancient growth temperatures during Asgard Archaea evolution. The authors resurrect many ancient EFs at various nodes of their phylogeny to conclude that the Asgard ancestor of eukaryotes was a moderate thermophile. The authors also characterize many extant EFs from Asgard to better understand growth temperature throughout the lineage. The manuscript presents interesting results that build off previous studies that describe ancient microbial life and thus makes a strong contribution despite the obvious contentious placement of eukaryotes within the Archaeal domain of life (a debate best left to other manuscripts). I am generally enthusiastic about the manuscript but this could be enhanced with additional clarity.

Response. We appreciate this positive assessment and the constructive comments which we address in detail below.

C1 (Comment 1). The title is a little misleading because optimal growth is not being directly measured. Maybe remove 'optimal'.

Response. The title was expanded to clarify what was actually done.

C2. The authors nicely describe how the ancient AncEF-1alpha sequence was inferred using multiple algorithms and then taking the consensus among those algorithm outputs. It is less clear if the ancestral Asgard nodes were inferred in a similar manner, or whether just one algorithm was used.

Response. The ancestral Asgard EF-1alpha nodes were inferred using three independent algorithms in a similar manner. We regret the lack of clarity in the original manuscript and have revised and expanded the description to clarify (lines 151–154).

C3. Figure 3 needs the nodes added from Figure S3. Please add filled-circles to all the nodes in Figure 3 in which an ancient protein was experimentally resurrected and characterized.

Response. The nodes and the related data have been added in Fig. 3 and Fig. S5.

C4. The authors mention that their results align with the results of a previous study that characterized ancient EF from ancestral alpha-proteobacteria. The authors also briefly mention this a limitation of dissolving molecular oxygen. I think readers would greatly benefit from an expanded discussion of these overlapping results and possible explanations for more complex life at this temperature transition.

Response. Thanks for the suggestion. We have revised and expanded the Discussion (e. g. lines 176–195) and hope it will be more informative for the readers.

C5. Minor points:

- Please add page numbers (always use page numbers).

Response. The page numbers have been added.

- P6, L120: 'with to the' is incorrect.

Response. Sorry, this mistake has been corrected.

- Ignorant point - I thought people started using eEF1A instead of EF-1alpha

Response. In the revised manuscript, we use EF-1A.

- Some of the temperature profiles in the Supp Material show broad profiles (and sometimes what appears to be a dual temperature optimum). Please discuss these unusual results. Experimental fluctuations, or possible properties from proteins that optimally function across a broad range of temperatures.

Response. To address this issue, we measured the thermal stability range of EF-1A in the temperature range from 0 to 100 °C, and further characterized the optimal GDP binding within the temperature range of protein stability. We found that some of the broad profiles became more narrow, and thus, we believe that most of the broad profiles resulted from experimental fluctuations (e. g. steps that remove free Mant-GDP by spin column). Although some broad profiles remained, they have little influence on our conclusion. The thermal stability range data of EF and updated GDP-binding results can be found in the manuscript and Fig. 1c, Fig. S3, S4, and S5.

Reviewer #2 (Remarks to the Author):

Evolution of optimum growth temperature in Asgard archaea – Lu et al.

In this manuscript Lu et al. measure the binding capacity of Mant-GDP to EF1 α proteins from extant Asgard species and from the inferred protein sequences of lost ancestors. They conclude that the last common ancestor of Asgard archaea had a preferred growth temperature of 65 °C and the Asgard ancestor of eukaryotes favored 50 °C. Such information is important for understanding the emergence of eukaryotes. However, there are a number of important issues to be addressed, particularly with the biochemical analysis, before publication should be considered.

Response. We appreciate the reviewer's interest and the general positive assessment of our work as well as the constructive comments that we address in detail below.

C1. The assay uses Mant-GDP to infer the optimal binding of GDP. A control sample is needed that verifies that Mant-GDP is an accurate reporter of GDP binding to EF1 α . To do this the authors can use an EF1 α from a bacterial species reported in reference 4 and relate the optimal binding to that publication. A second EF1 α control from a warm-blooded animal is needed to assess the relationship between Mant-GDP and a known EF1 α native environment. It would also be good to measure the optimal binding temperature for a second sequence from Fig. 2a, to demonstrate that small numbers of amino acid changes do not affect binding.

Response. Following the reviewer's suggestion, the EF proteins from *E. coli* (used in DOI: 10.1038/nature01977), *Homo sapiens*, *Saccharomyces cerevisiae* as well as the ancestral IQ-Anc

protein in Fig. 2a have been tested. The results can be found in the revised manuscript (lines 91–96 and 104–107), and Fig. S4 and S5.

C2. The assay in its current form is flawed. Most proteins denature below 100 °C. It is meaningless to analyze binding of Mant-GDP to denatured protein. Using these data points invalidate the statistical analysis. The authors should evaluate the stability range of each protein and only use/plot Mant-GDP data points that lie within the stable range. The ThermoFluor assay is one choice to measure the stability.

Response. We appreciate this suggestion. We have measured the thermal stability range of EF protein using ThermoFluor assay and adjusted the Mant-GDP binding assay accordingly. The data can be found in Fig. S3.

C3. The experimentally determined data points should replace the calculated plots in Fig. 1c and Fig. S3b,c. The current versions of these figures can be placed in the SD.

Response. Following this suggestion, the data points have been added to the Fig. 1c and Fig. S5. The statistical analysis of the optimal GDP-binding temperature can be found in Fig. S4.

C4. The relationships of the optimal binding temperatures to the reported sample collection environments for metagenomics need to be included for all extant species in this analysis. There seems to be large discrepancies, which bring the hypothesis into question. MKD1 was isolated from ocean sediment ~2 °C. The first Thorarchaeia and Hermodarchaeia from estuary and mangrove sediments, respectively. It is difficult to reconcile these environments with the optimal binding temperatures from the Mant-GDP assay of 23, 94, 85 °C, respectively. A thorough discussion of this issue is required.

Response. The results obtained with the control group (e. g. MK-D1 and *E. coli*) that are used in this and previous studies (DOI: 10.1038/nature01977; 10.1038/nature06510) suggest that the Mant-GDP data generally accurately reflect the OGT of prokaryotes. However, given that most ecological niches and the associated microbial communities are complex and dynamic, the discrepancy between sampling information and metagenomic result sometimes occurs. For example, it has been found that the optimal growth temperature of MK-D1 is around 20 °C (DOI: 10.1038/s41586-019-1916-6). It thus seems that association between the prokaryotic OGT (or GDP-binding temperature) and the sample collection environment might provide limited information. We address this issue in the revised manuscript (lines 206-214). Importantly, however, we corrected the OGT in addition to making the changes described above. In response to the specific comments of the reviewers, we also corrected the OGT values for Hermodarchaeia and Thorarchaeia to mesophilic values, having found that the high OGTs were caused by an unfortunate experimental error.

C5. The optimal binding temperature should be written on the branches and nodes in Fig. 3 and Fig S3a to aid the readers' understanding of the results.

Response. The temperature data has been added in Fig. 3 and Fig. S5.

Reviewer #3 (Remarks to the Author):

This manuscript by Lu et al. addresses the evolution of optimal growth temperatures across the Asgard archaea. To investigate this, the authors focus on the elongation factor 1alpha (EF-1alpha), a crucial component required for protein synthesis by the ribosome, whose affinity for GDP is optimal at a temperature that approximates that of optimal organismal growth. They reconstruct ancestral sequences for EF-1alpha across the phylogeny of the Archaea and experimentally test the optimal GDP-binding temperature of these ancestral sequences expressed in and purified from *E. coli*. The authors derive at least two broad conclusions: (1) there is somewhat of an overall macroevolutionary trend towards a decrease in OGT in Asgard archaea, and (2) the divergence of the eukaryotic nucleocytoplasmic lineage from Asgard archaea occurred at a point where these archaeal ancestors had become more mesophilic—the archaeal ancestor of eukaryotes was slightly thermophilic by being adapted to 50-60C.

Although this is an interesting study that addresses an important question, I have some methodological concerns and would caution against the broad conclusions that are stated. I thus think that this manuscript has to undergo major revisions before it can be published in a journal of broad readership.

Response. We appreciate the reviewer’s interest and the general positive assessment as well as the constructive comments that we address in detail below.

C1. It is most likely that the model used for phylogenetic inference and ancestral sequence reconstruction is not the best-fitting model as it does not account for cross-site heterogeneity in amino acid composition. IQ-TREE includes a set of mixture models (e.g., C10-60) that are particularly suited to deep phylogenetic issues like the one addressed here. The ‘TESTONLY’ option used by the authors would not test mixture models prior to phylogenetic analysis. This is a rather unfortunate choice of parameters.

Response. We appreciate this suggestion. We have reanalyzed the EF-1A phylogeny using the recommended best-fitting mixture model Q.pfam+C40+F+R6 in IQ-tree. The results indicate that the overall tree topologies obtained with the empirical and mixture models are similar (see the figure below). Compared with the mixture model tree, in the empirical model tree, Kariarchaeia are a sister group to the Gerdarchaeia-Hodarchaeia clade which is a topology that is better consistent with our universal tree (DOI: 10.1038/s41586-021-03494-3).

It is also noteworthy that in a recent study (DOI: 10.1002/pro.4393), phylogenetic analysis and ancestral sequence reconstruction of EF-1 were performed using the empirical model. Our resulting EF-1A phylogeny based on the empirical model can be compared to the results of this work.

Taken together, this evidence indicates that the empirical model is appropriate to infer the EF-1A phylogeny, and the resulting phylogeny is reliable. To make it clear, we have added the mixture model tree in Fig. S1 and revised the relevant description in the manuscript (lines 51–63).

C2. What is the effect of trimming on ancestral sequence reconstruction? Depending on the quality of the alignment and the stringency of the trimming, it could have serious effects. The authors need to explore this or at least provide a proper explanation.

Response. There are few studies that examine the effect of trimming on ancestral sequence reconstruction (e. g. DOI: 10.1038/nature01977; DOI: 10.1038/s41586-020-2292-y; DOI: 10.1038/nature10724; DOI: 10.1126/science.aay9959). In all likelihood, this effect varies across protein families.

In the EF-1A alignment used in this work, our primary sequence alignment of the EF proteins shows that G domain, Domain 1, and Domain 2 are highly conserved, but the terminal residues, that likely form loop structure in both ends of the EF protein, are distinct. Given their impact on GDP-binding structure is likely negligible, we removed these terminal residues and the trimmed EF were shown in Fig. S6. We have revised the description to make it clear (lines 230–233).

C3. The authors should also estimate the optimal growth temperature of the genomes analyzed using the genome-wide methods reported previously (<https://doi.org/10.1093/bioinformatics/btz059>). Ancestral reconstruction could then be made on these estimates using comparative phylogenetic methods, e.g., using models of continuous character evolution. These ancestral estimates should be compared to those obtained in this study. This is an important analysis because it constitutes an independent approach that may add or decrease support to their conclusions.

Response. We fully agree with the reviewer that inferring OGT based on genomic features is an important approach. Indeed, in a recent advanced study of Asgard evolution (DOI: <https://doi.org/10.1101/2023.03.07.531504>), Asgard OGTs were inferred using this approach. We thus used an alternative method (PMID: 31117361) to predict the Asgard OGTs (Table S2). We also performed a correlation analysis between our GDP-binding temperature and the inferred

Asgard OGTs (Fig. S7, lines 133-140). These data, together with the EF-1A assay, provide an important complementary insight into the evolution of OGT across the Asgard phylum. The work of Eme and colleagues on genomic feature-inferred Asgard OGT has been cited and discussed in the revised manuscript (lines 199–205).

C4. What is the precise relationship between optimal growth and GDP-binding temperatures across species? A scatter plot showing this correlation should be presented. Conclusions about point estimates should incorporate the errors derived from such cross-species phylogenetic correlations.

Response. As suggested by reviewer#2, we added more controls including EF-Tu of *Escherichia coli*, EF-1A proteins of *Saccharomyces cerevisiae* and *Homo sapiens*. We calculated Pearson correlation coefficient **R** between OGT and optimal GDP binding temperature from the data obtained with these controls as well as that of MK-D1. The result (Pearson correlation coefficient $r = 0.9906$) verified the strong correlation between the temperature optimum Mant-GDP binding of EF-1A and OGT (lines 91–98). Given such a strong correlation, we believe that incorporation of errors would have a limited if any impact on the conclusions.

C5. To properly provide insights into the temperature regime of eukaryogenesis, as the authors claim, the OGT of LECA has to be estimated as well—the authors only address the OGT of FECA in their study. This would provide a temperature range that would better constrain eukaryogenesis. However, for this to be properly done, the taxon sampling for eukaryotes in this study has to be improved considerably. The current taxon sampling is limited to a few animal, fungal, and plant species, in addition to *Trypanosoma brucei*. Please keep in mind that most of eukaryotic diversity is microbial and is currently distributed among several so-called ‘supergroups’.

Response. We appreciate this comment. Undoubtedly, exploring the LECA OGT is interesting and important. We are concerned, however, that even a substantially expanded eukaryotic taxon sampling would not help much because the overwhelming majority of the extant eukaryotes are mesophiles with a narrow range of OGT. Furthermore, eukaryogenesis was a highly complex evolutionary process that included at least one but possibly more events of endosymbiosis and the accompanying drastic transformation of the cellular organization (DOI: 10.1016/j.cub.2020.12.035; DOI: 10.1051/medsci/2022164; DOI: 10.1038/s41564-023-01378-y; DOI: 10.1038/s41564-020-0710-4). Thus, whereas we can infer the OGT of the last Asgard ancestor of eukaryotes with reasonable confidence, there is much less confidence about the FECA (which is not the same as the last Asgard ancestor but rather a complex cell resulting from endosymbiosis) and especially the LECA. A reasonable and important constraint seems to be the upper temperature limit for the eukaryotic life, ~60 °C (DOI: 10.1017/S1473550413000438; DOI: 10.1073/pnas.69.9.2426). We briefly address all these issues in the revised Discussion.

Given the uncertainties mentioned above, in this work, we chose to focus on the evolution of the OGT within Asgard. We hope that our study is a good start for investigating the temperature regime of eukaryogenesis, but undoubtedly, much more work remains to be done.

C6. Conclusions about ancestral OGTs chronologically matching the Huronian glaciation era are rather unconvincing in the absence of properly calibrated chronograms (at minimum add time to

Fig. 3). The argument is barely elaborated upon. This argument at least deserves a proper discussion of the literature.

Response. Following this suggestion, we have elaborated the argument (lines 215–224) which is hopefully clearer and more convincing.

C7. What do the authors mean by the following sentence? “The dissolved oxygen at this temperature condition could be necessary for the mitochondrial endosymbiosis occurred since the mitochondrial ancestor is thought to possess aerobic respiratory capability.” The meaning and implications of this statement are not clear at all. In fact, dissolved oxygen concentration increases with lower temperatures.

Response. We apologize for the lack of clarity in the original description. It is certainly true that the solubility of gases including oxygen increases at lower temperatures, and the Discussion has been revised accordingly (lines 184–186).

C8. Claims about temperature of mitochondrial ancestor needs to be critically revisited. Obvious biases are present in the study cited by the authors. The authors should probably estimate the OGT of the mitochondrial ancestor using the same methods employed here for Asgard archaea. That would add further support to the optimal temperature range at which eukaryogenesis occurred. Be sure to use the most up-to-date studied on the phylogenetic placement of the mitochondrial ancestor: <https://doi.org/10.1038/s41559-021-01638-2> and <https://doi.org/10.1038/s41586-018-0059-5>

Response. We appreciate this comment. Indeed, the phylogenetic placement of the mitochondrial ancestor remains problematic and may impact the OGT inference as we point out in the revision. Given that our work here focuses on the Asgard OGT evolution, exploring the OGT of the mitochondrial ancestor is beyond its scope. We have cited the key references (reference 28 and 29) and expanded the discussion in the manuscript (lines 191–195).

In addition to making the changes described above, in response to the specific comments of the reviewers, we corrected the OGT values for two classes of Asgardarchaeota, Hermodarchaeia and Thorarchaeia, because the high OGT assigned to these group ions the original manuscript.

Reviewer #1 (Remarks to the Author):

The authors have addressed all of the concerns/questions that I previously raised. I commend the other two reviewers for raising insightful concerns that allowed the authors to synthesize stronger results for an improvised article. The revised manuscript is a great piece of work and will be quite valuable to the research community.

Reviewer #2 (Remarks to the Author):

From Reviewer 2

The updated manuscript has addressed some of the issues with the original manuscript. However, two issues remain and a further point arises from the new Thermofluor data. Comments follow the numbering in the rebuttal.

Question C1 - resolved

This fully addresses the question. These controls have strengthened the manuscript.

Question C2 – partially resolved - reanalysis of data and text using the correct stability ranges is needed.

Figure S3, the thermal stability of each protein, improves the confidence of the GDP assays. However, there is a flaw in the manner in which these data are interpreted. The legend states that the peak corresponds to partial unfolding. I would argue that it is substantial, if not total, unfolding. Nonetheless, the intension of these data is to understand where the protein is stable. If we look at MK-D1 the point before the fluorescence signal increases (where the protein begins to unfold) it is 60 deg. Yet the authors have used binding data up to 80 deg (Fig. 1C) and state that it is stable up to 80 deg (line 89). Similarly, Fig. S3 shows that AncEF-1A is only stable up to 70 deg, yet again the binding data is reported to 80 deg (Fig 1D) and misreported in line 102. The statistical analysis of GDP binding (Fig. S4 and S5) is also using data beyond the point that these proteins are stable. All GDP-binding data, and calculations based on these data, need to be reanalyzed using the correct stability ranges for all proteins. The text and figures should be updated accordingly.

Question C3 – resolved

Question C4 – partially resolved

Yes, this is a complex issue. However, the manuscript gives the impression that the GDP-binding is giving a strong indication of ancestral habitat, which likely would also be a range. A clear statement that indicates that OGT values do not prescribe the exact environments of the ancestors is needed, rather that these organisms likely could have tolerated a range of temperatures. Here, the Thermofluor assay could perhaps be cited as a maximum limit beyond which the organisms would not survive.

In this section (around 209) line the MK-D1 example discussion can be expanded to comment on the Table S2 value.

Table S2 - MKD1 – OGT 31 deg

Table S1 optimal GDP-binding – 20 deg

Isolation source – 2 deg

Question C5 – resolved

New comment

The Thermofluor assay has shown that ancestral reconstructions produce artifacts - see <https://doi.org/10.1093/molbev/msw138>

Ancestors tend to be more stable than feasibly possible. The ancestors in this manuscript appear to be slightly more stable, in general. By analogy, the estimation of OGT via binding data may also have some bias from the reconstruction for the ancestors. A statement pointing out this uncertainty is needed.

Reviewer #3 (Remarks to the Author):

In their response to one of my points, the authors make a distinction between the "Asgard ancestor of eukaryotes" and FECA. Technically speaking, the "Asgard ancestor of eukaryotes" is the last common ancestor between eukaryotes and its closest sister Asgard archaeal group. While theoretically FECA and "Asgard ancestor of eukaryotes" are phylogenetically distinct entities, they are from a phenotypic standpoint practically identical. FECA is the first entity that can be considered part of total group Eukaryota, and as a consequence was just a generation away from the "Asgard ancestor of eukaryotes". FECA is not defined phenotypically as the authors are led to believe.

I am afraid these conceptual and terminological issues will have to be clarified before this manuscript is published. There has been way too much confusion for too long. A recent review makes these conceptual issues clear: <https://doi.org/10.1016/j.cub.2023.07.048>. But others have before as well, see for example: <https://doi.org/10.1038/nrmicro.2017.133>. The authors are referred to this literature.

L113-115: Most eukaryotes are mesophiles, and it is thus very likely that the ancestral reconstruction of the OGT of LECA would give a mesophilic temperature. However, the authors here claim that eukaryogenesis happened at a temperature range of 50-60C. This logic is flawed as the upper temperature limit for eukaryotic life does not help to constrain the temperature at which eukaryogenesis occurred. Perhaps what the authors want to say here is that eukaryogenesis may have started at a temperature of 60C.

L176-179: Why? The upper limit to eukaryotic life, namely about 60C, is a derived rather than an ancestral state. As said above, it is much more likely that LECA was a mesophile and thus the OGT decreased throughout eukaryogenesis from 50 to a much lower temperature.

L189-190: This statement is probably correct because it refers to the upper temperature limit of 60C, not to the range at which eukaryogenesis occurred.

L190-191: This sentence makes more sense and agrees with the point I was making before. This sentence thus conflicts with the one above by the same authors.

I think the authors should also at least estimate and discuss the inferred OGT of the "Asgard ancestor of eukaryotes" under an alternative topology supported by other studies where eukaryotes are sister to or within Heimdallarchaea. (Note that I am not asking for experiments of resurrected ancestral proteins, but the ancestral sequence reconstruction could be performed.)

REVIEWER COMMENTS

Reviewer #1 (Remarks to the Author):

The authors have addressed all of the concerns/questions that I previously raised. I commend the other two reviewers for raising insightful concerns that allowed the authors to synthesize stronger results for an improvised article. The revised manuscript is a great piece of work and will be quite valuable to the research community.

Response. We appreciate the reviewer's positive assessment of our work.

Reviewer #2 (Remarks to the Author):

From Reviewer 2

The updated manuscript has addressed some of the issues with the original manuscript. However, two issues remain and a further point arises from the new Thermofluor data. Comments follow the numbering in the rebuttal.

Response. We appreciate the reviewer's interest in our work as well as the constructive comments that we address in detail below.

Question C1 - resolved

This fully addresses the question. These controls have strengthened the manuscript.

Response. We appreciate the positive assessment.

Question C2 – partially resolved - reanalysis of data and text using the correct stability ranges is needed.

Figure S3, the thermal stability of each protein, improves the confidence of the GDP assays. However, there is a flaw in the manner in which these data are interpreted. The legend states that the peak corresponds to partial unfolding. I would argue that it is substantial, if not total, unfolding. Nonetheless, the intention of these data is to understand where the protein is stable. If we look at MK-D1 the point before the fluorescence signal increases (where the protein begins to unfold) it is 60 deg. Yet the authors have used binding data up to 80 deg (Fig. 1C) and state that it is stable up to 80 deg (line 89). Similarly, Fig. S3 shows that AncEF-1A is only stable up to 70 deg, yet again the binding data is reported to 80 deg (Fig 1D) and misreported in line 102. The statistical analysis of GDP binding (Fig. S4 and S5) is also using data beyond the point that these proteins are stable. All GDP-binding data, and calculations based on these data, need to be reanalyzed using the correct stability ranges for all proteins. The text and figures should be updated accordingly.

Response. We agree with the reviewer that the peak corresponds to the substantial protein unfolding in Fig. S3. We have reanalyzed the correct stability range for all proteins and revised the text (lines 89, 104, 519, and 520) and data (Fig. 1, Fig. S4, and Fig. S5) accordingly. Additionally, we have replotted the Fig. S3 using three original points to make it a little clearer.

Question C3 – resolved

Response. Thanks

Question C4 – partially resolved

Yes, this is a complex issue. However, the manuscript gives the impression that the GDP-binding is giving a strong indication of ancestral habitat, which likely would also be a range. A clear statement that indicates that OGT values do not prescribe the exact environments of the ancestors is needed, rather that these organisms likely could have tolerated a range of temperatures. Here, the Thermofluor assay could perhaps be cited as a maximum limit beyond which the organisms would not survive.

Response. We agree and, following the suggestion, we have added the statement that OGT do not prescribe the exact environment for ancestors (lines 223-225). We agree with the reviewer' s suggestion that the maximum thermostability limit of the EF-1A protein probably can be viewed as the upper temperature limit for the host and have added the respective statement (lines 91, 92).

In this section (around 209) line the MK-D1 example discussion can be expanded to comment on the Table S2 value.

Table S2 - MKD1 – OGT 31 deg

Table S1 optimal GDP-binding – 20 deg

Isolation source – 2 deg

Response. The statements have been added as suggested (lines 214-217).

Question C5 – resolved

Response. Thanks.

New comment

The Thermofluor assay has shown that ancestral reconstructions produce artifacts - see <https://doi.org/10.1093/molbev/msw138>

Ancestors tend to be more stable than feasibly possible. The ancestors in this manuscript appear to be slightly more stable, in general. By analogy, the estimation of OGT via binding data may also have some bias from the reconstruction for the ancestors. A statement pointing out this uncertainty is needed.

Response. Following the suggestion, we have added the statement (lines 221-223) and cited the key reference (reference 38).

Reviewer #3 (Remarks to the Author):

In their response to one of my points, the authors make a distinction between the “Asgard ancestor of eukaryotes” and FECA. Technically speaking, the “Asgard ancestor of eukaryotes” is the last common ancestor between eukaryotes and its closest sister Asgard archaeal group. While theoretically FECA and “Asgard ancestor of eukaryotes” are phylogenetically distinct entities, they are from a phenotypic standpoint practically identical. FECA is the first entity that can be considered part of total group Eukaryota, and as a consequence was just a generation away from the “Asgard ancestor of eukaryotes”. FECA is not defined phenotypically as the authors are led to believe.

I am afraid these conceptual and terminological issues will have to be clarified before this manuscript is published. There has been way too much confusion for too long. A recent review makes these conceptual issues clear: <https://doi.org/10.1016/j.cub.2023.07.048>. But others have before as well, see for example: <https://doi.org/10.1038/nrmicro.2017.133>. The authors are referred to this literature.

Response. We appreciate this important comment. We agree that the “last Asgard ancestor of eukaryotes” is phylogenetically defined as an archaeal lineage that was the last common ancestor between eukaryotes and its closest sister Asgard group. However, with all due respect, we cannot accept the definition of two FECAs (archaeal and bacterial) promoted in the recent review article cited by the reviewer (<https://doi.org/10.1016/j.cub.2023.07.048>). Indeed, this is not how FECA was defined in the first place (<https://pubmed.ncbi.nlm.nih.gov/16106042/>) or in other recent publications (<https://pubmed.ncbi.nlm.nih.gov/32341569/>; <https://pubmed.ncbi.nlm.nih.gov/37254790/>; <https://pubmed.ncbi.nlm.nih.gov/37127702/>). According to these definitions, FECA is the first true eukaryote, in all likelihood, already a symbiont between an Asgard archaeon and a bacterium. This is clearly distinct from the last Asgard ancestor of eukaryotes, both phylogenetically and biologically and is not just a generation away. We added these clarifications to the description of “the last Asgard ancestor of eukaryotes” and the FECA (lines 190-194) and cited the above references in the revised manuscript.

L113-115: Most eukaryotes are mesophiles, and it is thus very likely that the ancestral reconstruction of the OGT of LECA would give a mesophilic temperature. However, the authors here claim that eukaryogenesis happened at a temperature range of 50-60C. This logic is flawed as the upper temperature limit for eukaryotic life does not help to constrain the temperature at which eukaryogenesis occurred. Perhaps what the authors want to say here is that eukaryogenesis may have started at a temperature of 50C.

Response. The upper limit for eukaryotes helps to constrain eukaryogenesis from the high temperature side and is therefore relevant. However, we indeed infer the temperature of eukaryogenesis to be about 50C, and made this explicit in the revised manuscript (lines 115, 116).

L176-179: Why? The upper limit to eukaryotic life, namely about 60C, is a derived rather than an ancestral state. As said above, it is much more likely that LECA was a mesophile and thus the OGT decreased throughout eukaryogenesis from 50 to a much lower temperature.

Response. Yes, it is likely that LECA was a mesophile, and we say so explicitly in the revised manuscript (lines 196,197).

L189-190: This statement is probably correct because it refers to the upper temperature limit of 60C, not to the range at which eukaryogenesis occurred.

Response. No changes were made here.

L190-191: This sentence makes more sense and agrees with the point I was making before. This sentence thus conflicts with the one above by the same authors.

Response. No changes were made here.

I think the authors should also at least estimate and discuss the inferred OGT of the “Asgard ancestor

of eukaryotes” under an alternative topology supported by other studies where eukaryotes are sister to or within Heimdallarchaea. (Note that I am not asking for experiments of resurrected ancestral proteins, but the ancestral sequence reconstruction could be performed.)

Response. We are certainly well aware of alternative placements of Eukaryotes in the Asgard tree. However, for this work, we considered it prudent to use the tree of EF-1A itself which is one of the better phylogenetic markers and produces the topology in which eukaryotes are the sister group of the Heimdall-Kari-Gerd-Hod clade of Asgard archaea, with strong bootstrap support. We did not think that using an alternative topology that does not reflect the phylogenetic tree of the gene that is at the center of the current study was justified.

Reviewer #2 (Remarks to the Author):

My concerns have been fully addressed. I recommend publication without further delay.

Reviewer #3 (Remarks to the Author):

The authors refuse to accept that their definition of FECA is wrong. This is probably because one of the authors has previously written on the topic. However, this definition is not really a matter of differing opinions among scientists. The evolutionary, phylogenetic, and paleontological literature on how to slice a phylogenetic tree and name its entities is logical, clear, and uncontroversial. This means that previous definitions cited by the authors are simply wrong. Genomicists and biochemists who do not have formal training in evolutionary biology are unaware of this and often create new and conflicting definitions that only introduce confusion in the field.

The first ancestor of a group did not have the synapomorphies that are used to define the group. This is a direct consequence of understanding the evolutionary process and phylogenetic representation. Note also that their definition of FECA assumes a particular eukaryogenesis model (mitochondrion-first) which is by no means agreed upon by scientists in the field. This makes their FECA definition highly undesirable. Using a phylogenetic definition of FECA gets rid of this problem and makes it compatible with contrasting scenarios/models on the origin of eukaryotes. Note also that a phylogenetic definition of FECA is not something adopted by a particular faction of scientists with homogenous views—scientists with differing views on the origin of eukaryotes embrace it (as the citations demonstrate) simply because it is the only logical way to define FECA.

REVIEWER COMMENTS

Reviewer #2 (Remarks to the Author):

My concerns have been fully addressed. I recommend publication without further delay.

Response. We appreciate the positive assessment.

Reviewer #3 (Remarks to the Author):

The authors refuse to accept that their definition of FECA is wrong. This is probably because one of the authors has previously written on the topic. However, this definition is not really a matter of differing opinions among scientists. The evolutionary, phylogenetic, and paleontological literature on how to slice a phylogenetic tree and name its entities is logical, clear, and uncontroversial. This means that previous definitions cited by the authors are simply wrong. Genomicists and biochemists who do not have formal training in evolutionary biology are unaware of this and often create new and conflicting definitions that only introduce confusion in the field.

The first ancestor of a group did not have the synapomorphies that are used to define the group. This is a direct consequence of understanding the evolutionary process and phylogenetic representation. Note also that their definition of FECA assumes a particular eukaryogenesis model (mitochondrion-first) which is by no means agreed upon by scientists in the field. This makes their FECA definition highly undesirable. Using a phylogenetic definition of FECA gets rid of this problem and makes it compatible with contrasting scenarios/models on the origin of eukaryotes. Note also that a phylogenetic definition of FECA is not something adopted by a particular faction of scientists with homogenous views—scientists with differing views on the origin of eukaryotes embrace it (as the citations demonstrate) simply because it is the only logical way to define FECA.

Response. Following the editor's suggestion and given that the definition of FECA is not directly relevant to this work, we have removed the mention of FECA from the manuscript (lines 195-197).